# In Vitro Assessment of Long-Term Fluoride Ion Release from Nanofluorapatite

**DOI:** 10.3390/ma14133747

**Published:** 2021-07-04

**Authors:** Katarzyna Herman, Marta Wujczyk, Maciej Dobrzynski, Dorota Diakowska, Katarzyna Wiglusz, Rafal J. Wiglusz

**Affiliations:** 1Department of Pediatric Dentistry and Preclinical Dentistry, Wroclaw Medical University, Krakowska 26, 50-425 Wroclaw, Poland; katarzyna.herman@umed.wroc.pl; 2Institute of Low Temperature and Structure Research, Polish Academy of Sciences, Okolna 2, 50-422 Wroclaw, Poland; m.wujczyk@intibs.pl; 3Department of Nervous System Diseases, Wroclaw Medical University, Bartla 5, 51-618 Wroclaw, Poland; dorota.diakowska@umed.wroc.pl; 4Department of Analytical Chemistry, Wroclaw Medical University, Borowska 211 A, 50-566 Wroclaw, Poland; katarzyna.wiglusz@umed.wroc.pl

**Keywords:** fluorapatite, nanomaterials, tooth decay prophylaxis, fluoride ion release

## Abstract

The issue concerning the tooth decay is ongoing, therefore the study of materials with potential use in its prevention is crucial. This study aimed to analyze the long-term release of fluoride from synthesized nanofluorapatite in various in vitro environments for its potential use in dental materials. We placed 100 mg samples in 0.9% NaCl or deionized water and incubated them at 37 °C or 22 °C for 12 weeks. F− levels were read at 1, 3, 24, 48, 72, and 96 h, and thereafter weekly. The levels of F− released at specific time intervals, as well as their cumulative values were compared. In a solution of 0.9% NaCl at 22 °C, there were no significant differences in the amount of F− released in the assessed time intervals, while at 37 °C, the highest value was read after 24 h (0.0697 ppm + 0.0006; *p* < 0.05). In deionized water, the highest amount of F− at 22 °C was read after 4 weeks (0.0776 ppm + 0.0028; *p* < 0.05), and at 37 °C, it was also the highest after 4 weeks (0.0910 ppm + 0.0156; *p* < 0.05). Under the same conditions, after 5 weeks the cumulative level of F− released (0.6216 ppm + 0.0085) significantly increased (*p* < 0.05), when compared to the samples placed in 0.9% NaCl at 37 °C and 22 °C (0.5493 ppm + 0.0321 and 0.5376 ppm + 0.0234, respectively). FAp releases F− for a long period of time in all assessed environments, therefore it is advised to continue testing in vivo models. Due to the probable remineralization effect towards hard tooth tissues, fluorapatite can be used in the prevention and treatment of dental caries and dentin hypersensitivity.

## 1. Introduction

The concept of nanotechnology involves the use of a number of natural or synthetic materials with a particle size < 100 nm. The practical application of nanomaterials is found in many fields of science, including medicine [1]. Nanoparticles are used among other applications in the prevention of dental hard tissue diseases, conservative dentistry, prosthetics, periodontics or maxillofacial surgery [2]. Among other materials used in this area, attention should be given to apatites [3,4,5,6,7].

Regenerative medicine, but mostly of all, in the case of aesthetic dentistry and restoration of impaired hard tissue is of great importance [8,9]. Synthetic hydroxyapatite (HAp) has been thoroughly investigated for bone substitution, repair and tissue engineering based on its high biocompatibility [10,11,12,13,14,15,16,17]. Its chemical composition and crystal structure can be comparable to apatite found in bone tissue matrix [18]. HAp has found its application as a restorative material for orthopedics, dentistry and coating on implants due to its osteoconductive and osteoinductive potentials [14,19,20,21]. Due to the similarity of its chemical structure, hydroxyapatite has the ability to incorporate into the structure of tooth enamel. Therefore, it can be used in the prevention of caries and non-carious lesions, treatment of early carious lesions and reduction of dentin hypersensitivity [6,22].

When hydroxyl groups (OH^−^) are substituted by fluoride ions (F^−^), fluorapatite (FAp) is formed. Fluorapatite can be used as a complementary material for hydroxyapatite due to its better thermal stability, lower solubility, and comparable biocompatibility. The addition of hydroxyapatite or fluorapatite to glass ionomer cements (GIC) can improve the effectiveness of commercially sold dental cement, by improving its bond strength to dentin and mechanical properties, such as compressive strength, diametral tensile strength and hardness [4,23,24].

Fluorapatite and hydroxyapatite crystalize into the hexagonal crystal system, to the *P*6_3_/m space group. Unit cell dimensions of hydroxyapatite (*a* = 9.432 Å and *c* = 6.881 Å) are relatively larger than those of fluorapatite. Substitution by F^−^ ions does not change the atomic structure other than the *a*-axis dimensions to 9.368 Å, and the *c*-axis remains unchanged [25]. 

Fluorine is an essential microelement for normal maxilla and skeletal bone formation [26,27] as well as dental cavity prevention and repair [28]. The presence of fluorine ions may further positively influence the bone and teeth in the maxilla because fluorapatite can be found in tooth enamel [25].

Nowadays, it is believed that the constant, long-term, exogenous delivery of low fluoride concentrations to the oral cavity environment is the most beneficial in the prevention of caries [29,30]. It has been suggested that daily intake of fluorine should range between 1.5 and 4 mg/day, for dental caries to be effectively reduced [31]. The cariostatic mechanisms of fluoride include enamel demineralization inhibition, remineralization, and antibacterial activity promotion. The release of calcium and phosphate ions from the enamel, and thus the destruction of hydroxyapatite crystals, occurs when the pH in the oral cavity falls below the critical value (5.5). The incorporation of fluorine ions into the structure of enamel hydroxyapatite and their presence in the interprismatic space increases its resistance to acids. In a situation where the enamel has been demineralized, its repair is possible by re-embedding the lost ions. An increase in the pH of saliva is a necessary condition for this process to take place. In the presence of fluoride ions, remineralization occurs faster and at a lower pH. Additionally, the restored enamel becomes stronger. Fluoride has also been shown to disrupt carbohydrate metabolism in cariogenic bacteria [29]. 

Due to the benefits of the constant presence of fluoride in the oral cavity environment, it is recommended that hygiene products and dental materials should be enriched with its compounds. Thus, the products used in dentistry become a source of this element released into dental tissues and saliva [32,33,34,35,36]. Among other compounds, fluorapatites are utilized for such purposes.

The problem of fluoride release from synthetic apatites under experimental conditions is not widely discussed in the literature. Attempts have been made to analyze this issue after incorporating fluorapatite or hydroxyapatite into glass ionomer materials in deionized water environment [37,38]. The results obtained indicate a significant increase in the level of fluoride released from the material enriched with fluorapatite, with the highest level of release observed in the first days of the 4-week experiment [36]. However, it should be noted that the use of distilled water, even when heated to body temperature, does not reflect oral conditions. Therefore, the use of physiological saline could allow the experiment to be carried out in conditions closer to natural. It is also unclear how changes in temperature might affect the level of fluoride release. Promising results of the works of other authors, and at the same time little information on the discussed issue, prompted us to conduct an experiment with the use of a new nanofluorapatite, which could potentially be used in dental prophylaxis and conservative dentistry. The study aims to analyze the long-term release of fluorine ions from synthetic nanofluorapatite in various in vitro environments, in the context of its potential use in dental materials. Due to the long-term release of F^−^ ions from nanofluorapatite it can be used for the tooth decay prophylaxis.

## 2. Materials and Methods

### 2.1. Fluorapatite Synthesis

Nanocrystalline fluorapatite powders were synthesized via the co-precipitation method. The starting materials included calcium nitrate tetrahydrate (Ca(NO_3_)_2_·4H_2_O, ≥99% Acros Organics, Geel, Belgium), ammonium fluoride (NH_4_F, 98% Alfa Aesar, Haverhill, MA, USA), di-ammonium phosphate ((NH_4_)_2_HPO_4_, ≥98% Avantor Performance Materials, Gliwice, Poland) and ammonia solution for pH adjustment. The stoichiometric amounts of starting materials were dissolved in deionized water. Then, the solutions were mixed, and a synthesis was carried out on a magnetic stirring plate (Heidolph Instruments GmbH & CO. KG, Schwabach, Germany) at 100 °C for 1.5 h. The reaction was maintained at pH ~10, adjusted with aqueous ammonia. The precipitate obtained was washed and centrifuged until neutral pH was reached, but not less than three times. Finally, the materials were dried for 24 h at 70 °C and later heat-treated at 450 °C for 6 h to form crystallized nanoparticles. 

### 2.2. Physicochemical Analysis of Fluorapatite

The crystal phase purity of fluorapatite was analyzed via the X-ray diffraction (XRD) method. The obtained pattern was collected with an X’Pert PRO X-ray diffractometer (Cu Kα1, 1.54060Å) (PANalytical, Malvern Panalytical Ltd., Malvern, UK). The diffractogram was analyzed and assigned to the standard pattern from the Inorganic Crystal Structure Database (ICSD). Analysis of the morphology and size of fluorapatite powder was performed on the SEM (scanning electron microscope) FEI Nova NanoSEM 230 (Hillsboro, OR, USA) equipped with an energy-dispersive (EDS) spectrometer (EDAX Genesis XM4). The infrared spectrum measurement was performed using a Nicolet iS10 FT-IR (Waltham, MA, USA) spectrometer equipped with a HeNe laser as an infrared (IR) radiation source and an automated beam splitter exchange system (iS50 ABX containing a DLaTGSKBr detector) with a built-in all-reflective diamond ATR module (iS50 ATR), Thermo Scientific Polaris™. The spectrum was detected in the range of 400–1300 cm^−1^ on a potassium bromide (KBr) plate. 

### 2.3. Release Analysis of F^−^ Ions

Nanofluorapatite samples, weighing 100 mg each, were placed in 5 mL of saline or deionized water and left standing at 37 °C or 22 °C. The release of fluoride ions at specific time intervals was measured using an ORION 9609 ion-selective electrode (Thermo Fisher Scientific Co., Waltham, MA, USA) connected to a CPI-551 microcomputer. Saline or deionized water were changed after the measurements were taken. Measurements were taken at the following time intervals: 1, 3, 24, 48, 72, and 96 h, and thereafter weekly for up to 12 weeks from the beginning of the experiment. The levels of F− released at specific time intervals and their cumulative values were compared.

### 2.4. Statistical Analysis

All experiments were performed three times and descriptive data were expressed as the mean and standard deviation (±SD). One-way analyses of variance (ANOVAs) for independent groups (according to incubation conditions) or dependent samples (according to time periods) were used for comparisons of continuous data between more than two groups. Tukey’s test was used as a post-hoc analysis for intergroup comparisons. Pearson’s correlation coefficients (r) were calculated to evaluate associations between pairs of variables. A 2-tailed *p*-value of < 0.05 was considered statistically significant. Statistical analyses were conducted using Statistica v.13.3 (Tibco Software Inc., Palo Alto, CA, USA).

## 3. Results

### 3.1. Physicochemical Evaluation of Nanofluorapatite

The pure crystal phase of the nanocrystalline fluorapatite was affirmed via X-ray powder diffraction measurement (Figure 1). Obtained diffractogram was juxtaposed with a standard pattern from Inorganic Crystal Structure Database (No. 9444). No additional crystal phases were observed. The relative intensities of the diffraction peaks differ from the standard pattern, which can be attributed to the crystallographic texture. The main difference in the relative intensities is observed for the (002) and (004) planes. 

SEM images have been obtained for the fluorapatite (Figure 2). Elongated, nanosized and partially aggregated particles of fluorapatite can be distinguished. Based on the presented SEM images (Figure 2) histograms of the grain size distribution for two diameters length and width were determined. The obtained results allowed estimation of a representative, median particle size, 152 nm length and 56 nm width.

The Fourier transform infrared (FT-IR) spectrum was recorded covering the 400–1300 cm^−1^ spectral region at room temperature. Obtained spectrum consists of vibrations typical to the PO_4_^3−^ groups at 473 cm^−1^ (ν_2_), 568 cm^−1^ (ν_4_), 605 cm^−1^ (ν_4_), 965 cm^−1^ (ν_1_), 1039 cm^−1^ (ν_3_) and 1098 cm^−1^ (ν_3_) (Figure 3). The vibrational bands can be described as the non-degenerated stretching mode (ν_1_), and the triply degenerated asymmetric stretching mode (ν_3_) of P–O bonds. Additionally, observed vibrational bands are associated with the doubly degenerated (ν_2_) and the triply degenerated (ν_4_) bending modes of O–P–O bonds [39].

### 3.2. The In Vitro Release of F^−^ from Nanofluorapatite

The results of in vitro release of fluoride ions from FAp into saline solutions or deionized water in the selected time period were shown in Table 1 and Figure 4. ANOVA analysis for dependent samples showed statistically significant differences in the release of F^−^ ions in certain time periods in all tested incubation conditions (*p* < 0.05) (Table 1). In the solution of 0.9% NaCl and at 37 °C, significantly higher levels of F^−^ ions release were shown after 24 h and 48 h of incubation, and significantly lower–after 12 weeks (*p* < 0.05). In the solution of 0.9% NaCl and at 22 °C, the release of F^−^ ions was higher after 72 h and 504 h (3 weeks), but these results were not statistically significant. There was a significantly higher release of F^−^ ions after 504 h (3 weeks), 672 h (4 weeks) and 840 h (5 weeks) of incubation in deionized water and at 37 °C (*p* < 0.05). These values were the highest among the total obtained results (Figure 4). In the solution of deionized water and at 22 °C, the release of F^−^ ions was significantly higher after 672 h (4 weeks) of incubation (*p* < 0.05) (Table 1, Figure 4).

The lowest level of the release of F^−^ ions was observed after 1 h of incubation in H_2_O and at 37 °C (*p* < 0.05) while, in the same conditions, a significantly higher release of F^−^ ions was shown after 672 h (4 weeks) and 840 h (5 weeks) in comparison to 0.9% NaCl, 37 °C and 0.9% NaCl, 22 °C incubations (*p* < 0.05) (Table 1). 

We observed statistically significant negative correlations between selected time points and fluoride ion release in all tested incubation conditions (*p* < 0.05) (Figure 5A–D).

The cumulated concentration of F^−^ ions was significantly higher after 1 h and 3 h of incubation in NaCl, 37 °C and NaCl, 22 °C, and H_2_O, 22 °C than in H_2_O, 37 °C (*p* < 0.05) (Table 2). After 672 h (4 weeks) and 840 h (5 weeks), the release of fluoride ions significantly increased in H_2_O, 37 °C and H_2_O, 22 °C (*p* < 0.05) (Table 2). Finally, the highest cumulated level of F^−^ ions was observed during incubation in deionized water and at 37 °C (Table 2, Figure 6).

## 4. Discussion

In powder materials, the distribution of crystallographic orientations is not completely random and may form a texture, due to the nature of the preformed X-ray powder diffraction measurement. Therefore, the relative intensities of the diffraction peaks differ from the standard pattern. In this research, the main difference in relative intensities is observed for the (002) and (004) planes. Wang Y. et al. [40] showed that the (002) crystallographic texture may occur for hydroxyapatite coatings and with the occurrence of the (002) crystallographic texture, the intensity of the diffraction peak attributed to the (004) plane also increases. Due to the nature of the preformed X-ray powder diffraction measurement, a crystallographic texture can be observed.

The current concept of the etiology of dental caries assumes that one of the conditions for its development is the imbalance between the ongoing processes of enamel remineralization and demineralization. If demineralization is not inhibited at the molecular level, the structure of enamel and dentin is irreversibly damaged and requires supplementation with restorative materials. According to the current strategy of minimally invasive dentistry, the aim should be to restore the ionic balance as early as possible so that enamel damage can be completely remineralized without the need for surgical intervention. When necessary, the preparation of hard tissues should be as economical as possible, and the fillings should be biocompatible and strengthen dental tissues [41]. In order to meet the aforementioned assumptions, a number of preventive measures should be implemented. One of them is the use of remineralizing materials with the ability to repair damaged enamel at the molecular level by incorporating the lost calcium and phosphate ions. Due to the size of molecules and hydrophilic properties, nanohydroxyapatite exhibits a high remineralization potential and, therefore, can be used as an enamel reparative material [42]. It is also recommended that a low level of fluoride ions should be maintained in the oral cavity by using hygiene measures and long-term release materials. One of the available possibilities is the use of fluorinated hydroxyapatites or fluorapatites.

Clark et al. [43] covered prefabricated steel crowns with fluorhydroxyapatite, used for example in pediatric dentistry to rebuild significantly damaged milk teeth. The crowns were cemented onto extracted deciduous molars which were placed in gelatin at pH 4.3 for 9 weeks. The growth of apatite crystals, the release of fluorine ions and the reduction of enamel demineralization areas around the crown surface were found. Not only was the anti-caries effect achieved, but also the aesthetic conditions improved as the material was shown to be similar to the natural color of dental tissues. After 70 days of experiment, Lin et al. [44] observed that the release of fluorine ions from resin-modified glass ionomer cement enriched with nanofluorapatite or nanofluorhydroxyapatite was almost three times higher than in the control group. According to Malik et al. [37], fluorapatite incorporated into the structure of GIC (glass ionomer cement) significantly increases the release of F^−^ ions, in contrast to hydroxyapatite which does not show such properties.

The authors’ own research revealed the release of fluoride from FAp under all applied conditions throughout the experiment; however, there was a significant negative correlation between the duration of the experiment and the level of the released fluoride. With regard to time, environment and ambient temperature, significant differences in the amount of F^−^ ions released were also found. The greatest differences were observed in deionized water at 37 °C. The concentration of F^−^ ions was the lowest after one hour, but after 3, 4 and 5 weeks it reached the highest values. In the following weeks, there was a clear decrease in the release of F^−^ ions. Malik et al. [37] obtained different results, which under the same conditions found the highest level of fluoride released from the fluorapatite-enriched glass ionomer after the first hour of the experiment. Then the level gradually decreased and after 3 weeks of observation it was close to the level released from the glass ionomer itself. The profile and amount of fluoride released can therefore be influenced by both the structure of the applied fluorapatite and the material to which it is added. In own research after 5 weeks of the experiment in deionized water at 37 °C, the significantly highest cumulative level of released F^−^ ions could also be observed, when compared to both environments in 0.9% NaCl. Under these conditions, the total amount of released fluoride was the highest after 12 weeks. A similar release profile could be observed in deionized water at 22 °C. In the environment of 0.9% NaCl at 37 °C, the highest release of F^−^ occurred much earlier—after 24 and 48 h. By analyzing the relationship between the temperature of a given environment and the level of released F^−^, it was found that in deionized water placed at 37 ℃ compared to the same environment at 22 ℃, the level of F^−^ released was significantly lower at the beginning of the experiment, i.e., after the first hour and also after 3 h (in this case in relation to the cumulative value). No significant differences were observed in the following time intervals. In 0.9% NaCl environments, no significant differences in F^−^ release were found in dependence with the temperature.

A comparative analysis of the obtained results and those provided by other authors is difficult due to the fundamental differences in the chemical structure of the assessed compounds as well as different experimental conditions. There is an insufficient amount of scientific papers on the above issue in the available literature. Nevertheless, the research results are promising. However, it should be noted that the experiments carried out in laboratory conditions do not fully illustrate the mechanisms taking place in the environment of the oral cavity. Therefore, studies applied for in vitro models are insufficient to determine the specific use and behavior for in vivo model and the work should be continued.

The limitations of the research are certainly laboratory conditions, which do not fully reflect the conditions of the oral cavity environment. Therefore, this research should be treated as a pilot. In the next stage, it would be advisable to continue the investigation after incorporation into products used in restorative dentistry and dental prophylaxis for further in vitro and in vivo evaluation. Due to the potential remineralization properties, the assessed FAp may in the future be added to materials filling carious cavities of permanent and deciduous teeth (e.g., glass ionomer cements and composite materials). The long-term fluoride released from such material reduces the risk of secondary caries in the vicinity of the filling. As a component of toothpastes and other prophylactic as well as therapeutic preparations, FAp can increase the remineralizing effect and reduce dentin hypersensitivity.

## 5. Conclusions

The co-precipitation synthesis method obtained nanosized fluorapatite. The crystal phase purity of the nanofluorapatite was affirmed via XRD measurements. Additionally, a (002) crystallographic texture was observed. SEM images allowed the nanometric size of the particles to be determined. FT-IR spectra analysis of vibrational modes further reaffirmed the fluorapatite structure. The analyzed FAp showed the ability to release long-term F− under experimental conditions. The levels of the released ions differed significantly at specific time intervals and in compared environments. The greatest differences of released fluoride ions was observed in deionized water at 37 °C. Additionally the highest overall long-term release of F^−^ ions was also observed in deionized water at 37 °C. The release profiles in 0.9% NaCl and deionized water differed significantly. In the case of the 0.9% NaCl solution environment, the highest released amount of F^−^ ions was observed after 24 h—much earlier than in deionized water, but with comparatively lower concentrations than for deionized water after 5 weeks. It is advisable to continue research towards its practical use in dental materials especially for tooth decay prophylaxis.

## Figures and Tables

**Figure 1 materials-14-03747-f001:**
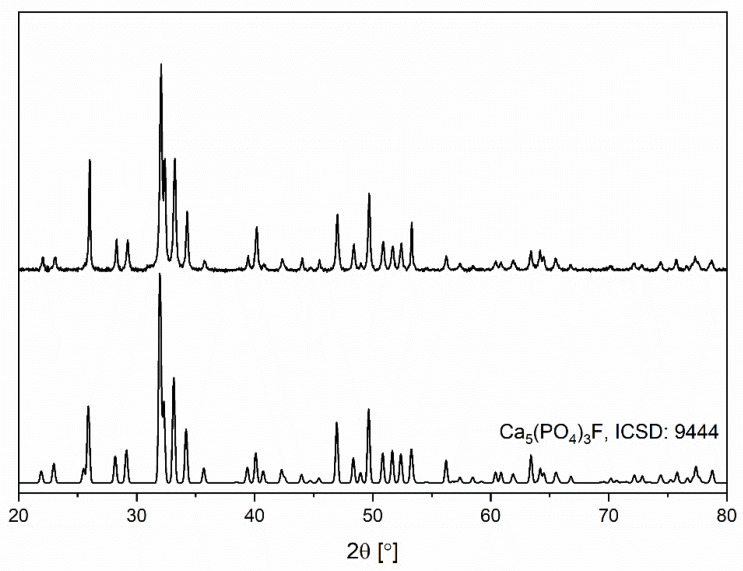
Powder diffractogram obtained for nanofluorapatite heat-treated at 450 °C for 6 h.

**Figure 2 materials-14-03747-f002:**
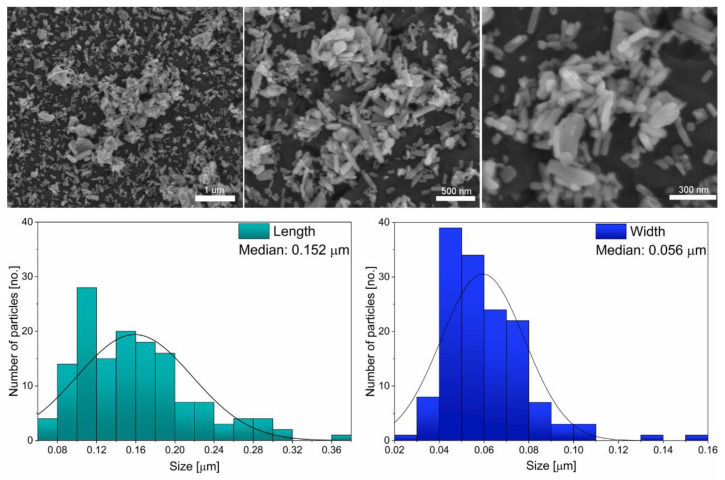
Scanning electron microscope (SEM) images (top) obtained for the investigated fluorapatite with its size distribution (bottom).

**Figure 3 materials-14-03747-f003:**
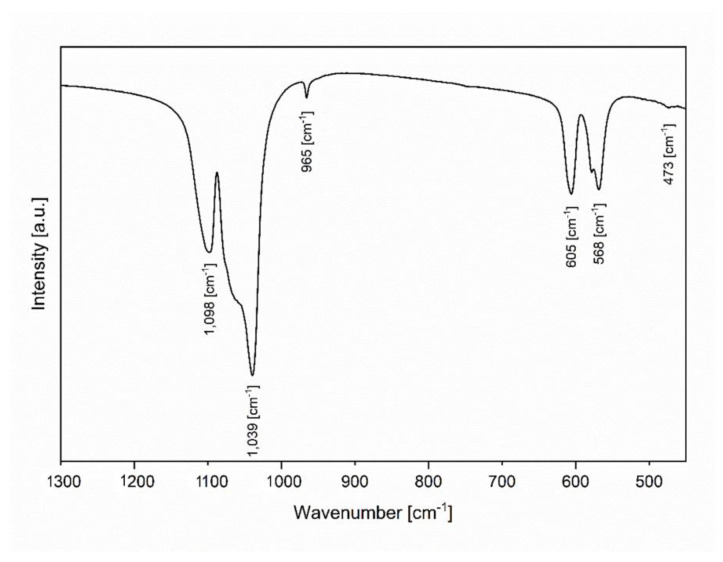
Fourier-transform infrared (FT-IR) spectrum of nanofluorapatite.

**Figure 4 materials-14-03747-f004:**
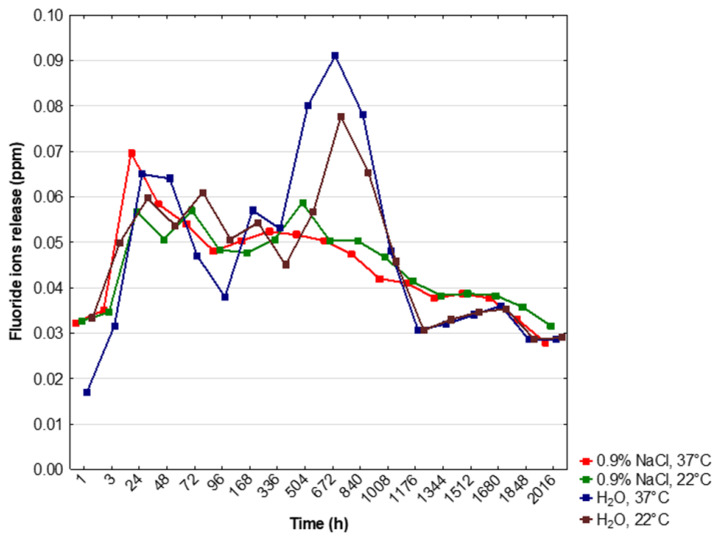
Relationship between time points and release of fluoride ions into a saline solution or deionized water. Dots represent the means of measurements.

**Figure 5 materials-14-03747-f005:**
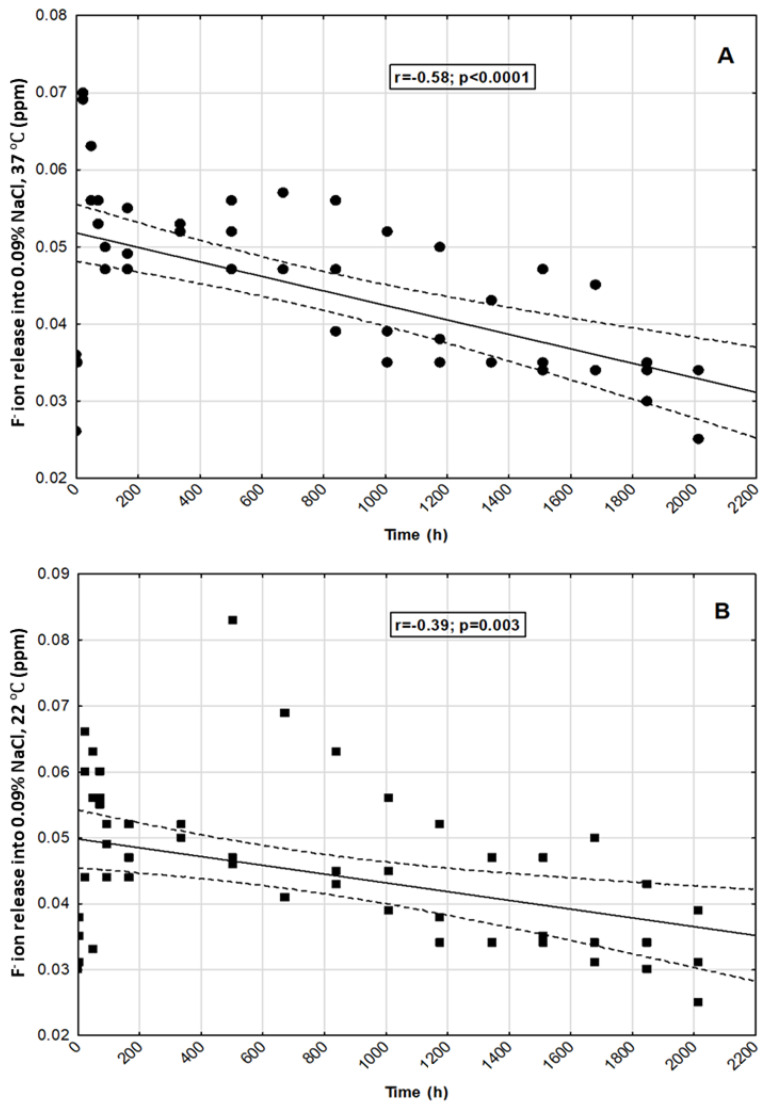
Correlation between selected time points and fluoride ion release into: (**A**)—0.9% NaCl, 37 °C; (**B**)—0.9% NaCl, 22 °C; (**C**)—H_2_O, 37 °C; (**D**)—H_2_O, 22 °C.

**Figure 6 materials-14-03747-f006:**
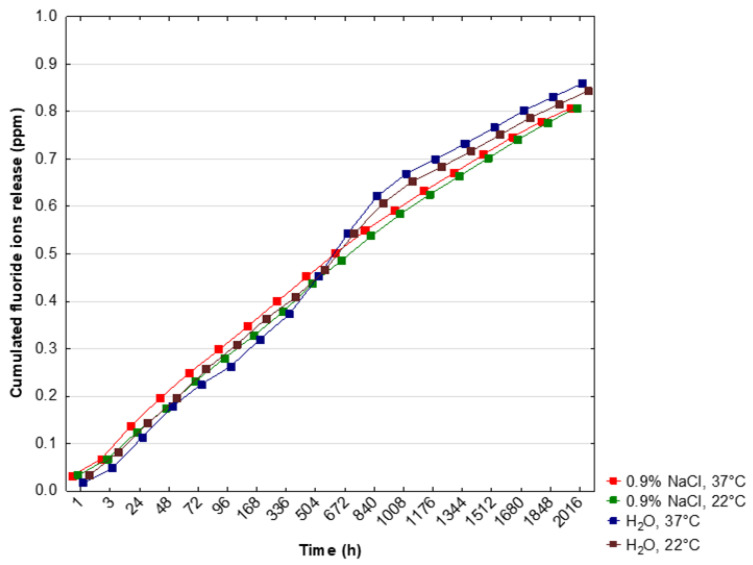
Cumulated fluoride ion release into a saline solution or deionized water. Dots represent the means of measurements.

**Table 1 materials-14-03747-t001:** The in vitro release of fluoride ions from nanofluorapatite into a saline solution or deionized water. Descriptive data were presented as the mean + standard deviation (+SD).

Time (Hour)	Fluoride Concentration (ppm)				*p*-Value (ANOVA for Independent Groups)
	0.9% NaCl, 37 °C [A]	0.9% NaCl, 22 °C [B]	H_2_O, 37 °C [C]	H_2_O, 22 °C [D]	
1 [1]	0.0323 ± 0.0055 ^#^	0.0326 ± 0.0046 ^#^	0.0170 ± 0.0026	0.0333 ± 0.0040 ^#^	0.004 *
3 [2]	0.0350 ± 0.0000 ^3^	0.0347 ± 0.0035	0.0316 ± 0. 0011	0.0500 ± 0.0141	0.057
24 [3]	0.0697 ± 0.0006 ^1^	0.0566 ± 0.0113	0.0650 ± 0.0034 ^1,2^	0.0596 ± 0.0075	0.191
48 [4]	0.0583 ± 0.0040 ^1^	0.0506 ± 0.0156	0.0640 ± 0.0017 ^1,2^	0.0536 ± 0.0196	0.619
72 [5]	0.0540 ± 0.0017	0.0570 ± 0.0026	0.0470 ± 0.0138 ^1^	0.0610 ± 0.0091	0.295
96 [6]	0.0480 ± 0.0017	0.0483 ± 0.0040	0.0380 ± 0.0121	0.0506 ± 0.0011	0.157
168 (1 week) [7]	0.0503 ± 0.0041	0.0476 ± 0.0040	0.0570 ± 0.0026 ^1^	0.0543 ± 0.0075	0.175
336 (2 weeks) [8]	0.0523 ± 0.0006	0.0506 ± 0.0011	0.0530 ± 0.0000 ^1^	0.0450 ± 0.0170	0.667
504 (3 weeks) [9]	0.0516 ± 0.0045	0.0586 ± 0.0210	0.0800 ± 0.0036 ^1,2^	0.0566 ± 0.0231	0.215
672 (4 weeks) [10]	0.0503 ± 0.0057 ^#^	0.0503 ± 0.0161 ^#^	0.0910 ± 0.0156 ^1,2^	0.0776 ± 0.0028	0.005 *
840 (5 weeks) [11]	0.0473 ± 0.0085 ^#^	0.0503 ± 0.0110 ^#^	0.0780 ± 0.0070 ^1,2^	0.0653 ± 0.0056	0.006 *
1008 (6 weeks) [12]	0.0420 ± 0.0088 ^3^	0.0466 ± 0.0086	0.0480 ± 0.0026 ^1,9,10,11^	0.0456 ± 0.0070	0.772
1176 (7 weeks) [13]	0.0410 ± 0.0079 ^3^	0.0413 ± 0.0094	0.0306 ± 0.0006 ^3,4,9,10,11^	0.0306 ± 0.0045 ^10^	0.129
1344 (8 weeks) [14]	0.0376 ± 0.0046 ^3^	0.0383 ± 0.0075	0.0320 ± 0.0017 ^3,4,9, 10,11^	0.0330 ± 0.0026	0.308
1512 (9 weeks) [15]	0.0386 ± 0.0072 ^3^	0.0386 ± 0.0072	0.0340 ± 0.0000 ^3,4,9, 10,11^	0.0346 ± 0.0040	0.616
1680 (10 weeks) [16]	0.0376 ± 0.0063 ^3^	0.0383 ± 0.0102	0.0360 ± 0.0017 ^9,10,11^	0.0353 ± 0.0023	0.924
1848 (11 weeks) [17]	0.0330 ± 0.0026 ^3,4^	0.0356 ± 0.0066	0.0286 ± 0.0023 ^3,4,9,10,11^	0.0286 ± 0.0023 ^10^	0.154
2016 (12 weeks) [18]	0.0280 ± 0.0052 ^3,4,8^	0.0316 ± 0.0070	0.0286 ± 0.0046 ^3,4,9,10,11^	0.0290 ± 0.0034 ^10^	0.835
***p*-value (ANOVA for dependent samples)**	<0.0001 *	0.0167 *	<0.0001 *	<0.0001 *	

*: statistically significant; ^#^: *p* < 0.05 vs. [C] in post-hoc test; NS: non-significant; ^1^: *p* < 0.05 vs. [1]; ^2^: *p* < 0.05 vs. [2]; ^3^: *p* < 0.05 vs. [3]; ^4^: *p* < 0.05 vs. [4]; ^8^: *p* < 0.05 vs. [8]; ^9^: *p* < 0.05 vs. [9]; ^10^: *p* < 0.05 vs. [10]; ^11^: *p* < 0.05 vs. [11].

**Table 2 materials-14-03747-t002:** Cumulated in vitro release of fluoride ions from nanofluorapatite into a saline solution or deionized water. Descriptive data were presented as the mean ± standard deviation (±SD).

Time (Hour)	Fluoride Concentration (ppm)				*p*-Value (ANOVA for Independent Groups)
	0.9% NaCl, 37 °C [A]	0.9% NaCl, 22 °C [B]	H_2_O, 37 °C [C]	H_2_O, 22 °C [D]	
1	0.0323 + 0.0055 ^###^	0.0326 ± 0.0046 ^###^	0.0170 ± 0.0026	0.0333 ± 0.0040 ^###^	0.004 *
3	0.0673 ± 0.0055 ^###^	0.0673 ± 0.0020 ^###^	0.0486 ± 0.0037	0.0833 ± 0.0105 ^###^	0.001 *
24	0.1370 ± 0.0051	0.1240 ± 0.0115	0.1136 ± 0.0032	0.1430 ± 0.0173	0.041
48	0.1953 ± 0.0080	0.1746 ± 0.0270	0.1776 ± 0.0041	0.1966 ± 0.0366	0.557
72	0.2493 ± 0.0095	0.2316 ± 0.0247	0.2246 ± 0.0113	0.2576 ± 0.0457	0.457
96	0.2973 ± 0.0110	0.2800 ± 0.0288	0.2626 ± 0.0186	0.3083 ± 0.0465	0.314
168 (1 week)	0.3476 ± 0.0145	0.3276 ± 0.0327	0.3196 ± 0.0185	0.3626 ± 0.0392	0.302
336 (2 weeks)	0.4000 ± 0.0151	0.3783 ± 0.0336	0.3726 ± 0.0185	0.4076 ± 0.0221	0.276
504 (3 weeks)	0.4516 ± 0.0185	0.4370 ± 0.0173	0.4526 ± 0.0170	0.4643 ± 0.0224	0.424
672 (4 weeks)	0.5020 ± 0.0242	0.4873 ± 0.0159	0.5436 ± 0.0015 ^##^	0.5420 ± 0.0020 ^##^	0.011 *
840 (5 weeks)	0.5493 ± 0.0321	0.5376 ± 0.0234	0.6216 ± 0.0085 ^#, ##^	0.6073 ± 0.0261 ^##^	0.006 *
1008 (6 weeks)	0.5913 ± 0.0410	0.5843 ± 0.0316	0.6696 ± 0.0100	0.6530 ± 0.0314	0.020 * post-hoc test: NS
1176 (7 weeks)	0.6323 ± 0.0352	0.6256 ± 0.0405	0.7003 ± 0.0100	0.6836 ± 0.0349	0.054
1344 (8 weeks)	0.6700 ± 0.0398	0.0664 ± 0.0471	0.7323 ± 0.0102	0.7166 ± 0.0374	0.125
1512 (9 weeks)	0.7086 ± 0.0370	0.7026 ± 0.0535	0.7663 ± 0.0102	0.7513 ± 0.0402	0.195
1680 (10 weeks)	0.7463 ± 0.0433	0.7410 ± 0.0635	0.8023 ± 0.0107	0.7866 ± 0.0413	0.310
1848 (11 weeks)	0.7793 ± 0.0453	0.7766 ± 0.0701	0.8310 ± 0.0101	0.8153 ± 0.0436	0.455
2016 (12 weeks)	0.8073 ± 0.0505	0.8083 ± 0.0771	0.8596 ± 0.0118	0.8443 ± 0.0470	0.546

*: statistically significant; NS: non-significant; ^#^: *p* < 0.05 vs. [A] in post-hoc test; ^##^: *p* < 0.05 vs. [B] in post-hoc test; ^###^: *p* < 0.05 vs. [C] in post-hoc test.

## Data Availability

Not applicable.

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
