# Peer review of "In Vitro Assessment of Long-Term Fluoride Ion Release from Nanofluorapatite"

_materials, 2021, doi:10.3390/ma14133747_

Round 1

Reviewer 1 Report

The manuscript reports the fluoride ion release performance of nanofluorapatite. By carrying experimental measurements, the authors claim that nanofluorapatite could release fluoride ion on a long-term basis in both deionized water and sodium chloride (NaCl) solution. While the fluoride release is a vital topic for both restorative and preventive dentistry, the manuscript does not provide a full insight into the results they obtained from experimental investigations (Please see the comments for the authors for specific details). Therefore, this manuscript is not suitable for publication in Materials.

Author Response

Dear Editor,

We would like to express our sincerest gratitude to the Reviewers for their enormous efforts in criticizing the manuscript. We have taken into account all raised question here follows the detailed answers to your as well as the Reviewers. Moreover, all changes we have made to the original manuscript and marked in the red colour in the text.

Reviewer 1:

The manuscript reports the fluoride ion release performance of nanofluorapatite. By carrying experimental measurements, the authors claim that nanofluorapatite could release fluoride ion on a long-term basis in both deionized water and sodium chloride (NaCl) solution. While the fluoride release is a vital topic for both restorative and preventive dentistry, the manuscript does not provide a full insight into the results they obtained from experimental investigations (Please see the comments for the authors for specific details). Therefore, this manuscript is not suitable for publication in Materials.

Answer: we do not see additional comments made by the Reviewer in the Comments and Suggestions for Authors section other than these stated above. Therefore, we will further address only the issues cited before. We present the data obtained for the fluoride ions release in Table 1 and cumulative values in Table 2. Additionally, release vs time correlations were presented in the form of figures (Figure 4, 5[A, B, C, D], 6).  Presented data, especially in tables 1 and 2, gives an information on the specific values obtained during the measurement over the span of 12 weeks.

Moreover, in our study we used a simplified model of the release of fluoride ions from fluorapatite into demineralized water and saline solution contains 0.9 percent sodium chloride. The solutions simulated body fluid, such as the Hank's physiological solution, the artificial saliva, the Dulbecco's phosphate-buffered saline, imitate physiological conditions and can be used for the release of substances. However, the presence of calcium and magnesium ions, or phosphate, carbonate anions in them causes the overlapping of the fluorine release process and the precipitation reaction of poorly soluble fluoride compounds[1]. As it was described in [2], OH- groups can be replaced by F- ions in the apatite structure and leads to a change in the solubility of apatite. In the structure of hydroxyapatite, phosphate ions can be exchanged for carbonate ions. Substitutions in the apatite by other chloride ions are also possible, and calcium ions can be substituted by sodium, potassium, or other elements[3]. Additionally, dissolving the fluorapatite is a source of the additional amounts of ions phosphate and calcium.

Therefore, a simple model with minimum of variables is beneficial for understanding and presenting the results of F- release from FAP, without imposing the additional processes.  On the other hand, the research design with the human liquids model is a very interesting and may be the next step in our experiment. Furthermore, the SEM images and the grain size distribution analysis has been added and described in the manuscript. Additionally, the elemental composition was checked by the means of EDS spectroscopy:

Calcium

26.3 %av

Phosphorus

12.5 %av

Oxygen

57.0 %av

Fluorine

4.2 %av

[1] Hattab FN, Amin WM. Fluoride release from glass ionomer restorative materials and the effects of surface coating. Biomaterials. 2001, 22(12):1449–58

[2] Fluorine and Health: Molecular Imaging, Biomedical Materials and Pharmaceuticals

Edited by Alain Tressaud, Gunter Haufe, Germany, 2008, 296-300

[3] E.A.Belousovaa,W.L.GriffinabSuzanne, Y.O'Reillya, N.I.Fishera Apatite as an indicator mineral for mineral exploration: trace-element compositions and their relationship to host rock type Journal of Geochemical Exploration, 2002, 76, 45-69

Reviewer 2 Report

  1. I) I don't feel that this work generates any new information or offers new thoughts or opinion. There is no explanation of significant novelty of the work as compared with previous works. Therefore, the authors need to explain the fundamental difference between their work and all previous studies.
  2. II) The introduction should be dedicated to present critical analysis of state of the art related work to justify the objective of the study. In this case, overall, the introduction section is too simple without a comprehensive description. Some content might be removed (information not directly related to the manuscript) and more content added, such as, whether previous studies have carried out, what is the weakness of previous studies, and why this study is meaningful and necessary.

III) Materials and Methods chapter – Materials and methods were insufficiently delineated. Please describe equipment used in the experiment – work development environment / work apparatus should be given – model of equipment (manufacturer, city, country). Also, an elemental analysis in order to confirm the chemical composition of nanofluoroapatite shoud be introduced.

  1. IV) Results and discussion. The term „nanofluoroapatite” is not suported by the obtained results. The authors should prepare TEM images for detailed investigation of structure, morphology and size of samples. Also, to increase the scientific value of the manuscript, the authors should consider extension of the all results section with comparison of obtained results with the results described in previous publications.
  2. V) The conclusions section needs to improve with selected and highlighted main findings.
  3. VI) English of the paper is rather good – in my opinion the language of the paper should be a little improved. I am asking for corrections by a native speaker.

On this basis, I believe that the present manuscript is not suitable for its publication in a high ranking journal such as Materials, in the present form. If the manuscript will not be significant improved, I will not recommend this paper for publication. I would be happy to review a revision of the manuscript that considers all my comments.

Author Response

Dear Editor,

We would like to express our sincerest gratitude to the Reviewers for their enormous efforts in criticizing the manuscript. We have taken into account all raised question here follows the detailed answers to your as well as the Reviewers. Moreover, all changes we have made to the original manuscript and marked in the red colour in the text.

Reviewer 2:

On this basis, I believe that the present manuscript is not suitable for its publication in a high ranking journal such as Materials, in the present form. If the manuscript will not be significant improved, I will not recommend this paper for publication. I would be happy to review a revision of the manuscript that considers all my comments.

Question 1:

I don't feel that this work generates any new information or offers new thoughts or opinion. There is no explanation of significant novelty of the work as compared with previous works. Therefore, the authors need to explain the fundamental difference between their work and all previous studies.

Answer: The manuscript describes the ability of long-term release of fluoride ions from synthesized via co-precipitation method nanofluorapatite up to 12 weeks. The long-term release of F- from the nanosized fluorapatite shows its novelty in the type of applied material as well as the specific various environments. Works detailing the release of fluoride ions concern mainly the release from the materials such as: ormocer materials[1], bioactive glass powders[2], modified glass ionomer cement (GIC)[3],[4],[5]. There are available works concerning release of F- from materials mainly being composites with fluorapatite, but not the fluorapatite itself. Manuscript has additional value as a comparative basis for composite materials using fluorapatite, taking into account the degree of fluoride release from the fluorapatite itself in various in vitro environments. The release of fluoride is considered in various in vitro environments, deionized water, and saline solution. The simplified solutions were chosen based on the available literature data. This approach to conducting research led to obtaining in our opinion novel results, not influenced release by the solutions simulated body fluid. The presence of calcium and magnesium ions, or phosphate, carbonate anions in them causes the overlapping of the fluorine release process and the precipitation reaction of poorly soluble fluoride compounds[6]. Therefore, a simple model with minimum of variables is beneficial for understanding and presenting the results of F- release from FAp, without imposing the additional processes.

Question 2:

The introduction should be dedicated to present critical analysis of state of the art related work to justify the objective of the study. In this case, overall, the introduction section is too simple without a comprehensive description. Some content might be removed (information not directly related to the manuscript) and more content added, such as, whether previous studies have carried out, what is the weakness of previous studies, and why this study is meaningful and necessary.

Answer: Information has been added to the manuscript.

Question 3:

Materials and Methods chapter – Materials and methods were insufficiently delineated. Please describe equipment used in the experiment – work development environment / work apparatus should be given – model of equipment (manufacturer, city, country). Also, an elemental analysis in order to confirm the chemical composition of nanofluoroapatite shoud be introduced.

Answer: The equipment was further described in materials and methods section. Additionally, we would like to present the results on elemental compositions obtained by the means of EDS spectroscopy:

Calcium

26.3 %av

Phosphorus

12.5 %av

Oxygen

57.0 %av

Fluorine

4.2 %av

Question 4:

Results and discussion. The term „nanofluoroapatite” is not suported by the obtained results. The authors should prepare TEM images for detailed investigation of structure, morphology and size of samples. Also, to increase the scientific value of the manuscript, the authors should consider extension of the all results section with comparison of obtained results with the results described in previous publications.

Answer: SEM images have been prepared for the size and morphology determination of investigated fluorapatite. Information has been added to the manuscript. Due to the lack of publications in which the experiment would be conducted under similar conditions, it is difficult to make a broad comparative analysis. In a few studies on this issue, different types of apatites have been used, combining them with commercially available dental materials. Different environments and experimental periods were also used.

Question 5:

The conclusions section needs to improve with selected and highlighted main findings.

Answer: The conclusions were changed, and the highlighted findings were added to the section.

Question 7:

English of the paper is rather good – in my opinion the language of the paper should be a little improved. I am asking for corrections by a native speaker.

Answer: The article underwent a linguistic proofreading before it was sent to the journal. The proofreading was performed by an external company dealing with translations into English.

[1] Kosior, P.; Dobrzynski, M.; Zakrzewska, A.; Grosman, L.; Korczynski, M.; Blicharski, T.; Gutbier, M.; Watras, A.; Wiglusz, R.J. Preliminary In Vitro Study of Fluoride Release from Selected Ormocer Materials. Materials 2021, 14, 2244.

[2] Gul, H.; Zahid, S.; Zahid, S.; Kaleem, M.; Khan, A.S.; Shah, Asma, T.S. Sol-gel derived fluoride-doped bioactive glass powders: Structural and long-term fluoride release/pH analysis, Journal of Non-Crystalline Solids. Volume 498, 15 October 2018, Pages 216-222

[3] Moshaverinia, M.; Borzabadi-Farahani, A.; Sameni, A.; Moshaverinia, A.; Ansari, S. Effects of incorporation of nano-fluorapatite particles on microhardness, fluoride releasing properties, and biocompatibility of a conventional glass ionomer cement (GIC). Dental Materials Journal 2016; 35(5): 817–821

[4] De Witte, A.M.J.C.; De Maeyer, E.A.P.; Verbeeck, R.M.H.; Martens, L.C. Fluoride release profiles of mature restorative glass ionomer cements after fluoride application. Biomaterials 21 (2000) 475-482

[5] Williams, J.A.; Billington, R.W.; Pearson, G.J. A long-term study of fluoride release from metal-containing conventional and resin-modified glass-ionomer cements. Journal of Oral Rehabilitation 28 (2001), 41-47

[6] Hattab FN, Amin WM. Fluoride release from glass ionomer restorative materials and the effects of surface coating. Biomaterials. 2001, 22(12):1449–58

Reviewer 3 Report

The paper “ In vitro assessment of long-term fluoride ion release from nanofluorapatite” presented the data on the fluoride release in distillate water and 0.9 % NaCl solution. The phase composition and FTIR spectra also were investigated. Unfortunately, the is no data on the chemical composition and morphology of the obtained nanopowders. The amount of the introduced fluorine ions influences its release in the solutions as well as powder’s morphology, particle sizes, and specific surface area. The investigation on the distillate water could not be considered as the model for human liquids, and there are a lot of more suitable solutions, for example, Hank’s physiological solution, artificial saliva, simulated body fluid, Dulbecco's phosphate-buffered saline. The additional investigations could significantly improve the presented paper. 

Author Response

Dear Editor,

We would like to express our sincerest gratitude to the Reviewers for their enormous efforts in criticizing the manuscript. We have taken into account all raised question here follows the detailed answers to your as well as the Reviewers. Moreover, all changes we have made to the original manuscript and marked in the red colour in the text.

Reviewer 3:

The paper “ In vitro assessment of long-term fluoride ion release from nanofluorapatite” presented the data on the fluoride release in distillate water and 0.9 % NaCl solution.

Question 1:

The phase composition and FTIR spectra also were investigated. Unfortunately, the is no data on the chemical composition and morphology of the obtained nanopowders. The amount of the introduced fluorine ions influences its release in the solutions as well as powder’s morphology, particle sizes, and specific surface area.

Answer: SEM images and the grain size distribution analysis has been added and described in the manuscript. Additionally, the elemental composition was checked by the means of EDS spectroscopy:

Calcium

26.3 %av

Phosphorus

12.5 %av

Oxygen

57.0 %av

Fluorine

4.2 %av

Question 2: 

The investigation on the distillate water could not be considered as the model for human liquids, and there are a lot of more suitable solutions, for example, Hank’s physiological solution, artificial saliva, simulated body fluid, Dulbecco's phosphate-buffered saline. The additional investigations could significantly improve the presented paper. 

Answer: In our study we used a simplified model of the release of fluoride ions from fluorapatite into demineralized water and saline solution contains 0.9 % sodium chloride. The solutions simulated body fluid, such as the Hank's physiological solution, the artificial saliva, the Dulbecco's phosphate-buffered saline, imitate physiological conditions and can be used for the release of substances. However, the presence of calcium and magnesium ions, or phosphate, carbonate anions in them causes the overlapping of the fluorine release process and the precipitation reaction of poorly soluble fluoride compounds[1]. As it was described in [2], OH- groups can be replaced by F- ions in the apatite structure and leads to a change in the solubility of apatite. In the structure of hydroxyapatite, phosphate ions can be exchanged for carbonate ions. Substitutions in the apatite by other chloride ions are also possible, and calcium ions can be substituted by sodium, potassium, or other elements[3]. Additionally, dissolving the fluorapatite is a source of the additional amounts of ions phosphate and calcium.

Therefore, a simple model with minimum of variables is beneficial for understanding and presenting the results of F- release from FAP, without imposing the additional processes. On the other hand, the research design with the human liquids model is a very interesting and may be the next step in our experiment.

[1] Hattab FN, Amin WM. Fluoride release from glass ionomer restorative materials and the effects of surface coating. Biomaterials. 2001, 22(12):1449–58.

[2] Fluorine and Health: Molecular Imaging, Biomedical Materials and Pharmaceuticals.

Edited by Alain Tressaud, Gunter Haufe, Germany, 2008, 296-300

[3] E.A.Belousovaa,W.L.GriffinabSuzanne, Y.O'Reillya, N.I.Fishera Apatite as an indicator mineral for mineral exploration: trace-element compositions and their relationship to host rock type Journal of Geochemical Exploration, 2002, 76, 45-69.

Reviewer 4 Report

This is a very interesting study on the release of fluoride ion from nanofluorapatite
I congratulate the authors for the design and implementation of the studies.
Some criticisms are however present:
- An opening sentence in the abstract section on the general problem must be inserted
-Line 24 A final sentence in the abstract section on the possible effects and clinical implications of the study and its results should be added
-In the introduction section it is necessary to add to the possible fields of application of hydroxyapatite also the possibility, highlighted in the literature, of its use as a filler for restorative materials. In this regard, in addition to the works entered by the authors, I recommend that you include the following scientific work in the reference section, which could be of help to the reader:
Chieruzzi, M .; Pagano, S .; Lombardo, G .; Marinucci, L .; Kenny, J.M .; Torre, L .; Cianetti, S. Effect of
nanohydroxyapatite, antibiotic, and mucosal defensive agent on the mechanical and thermal properties of
glass ionomer cements for special needs patients. J. Mater. Res. 2018, 33, 638–649.
- At the end of the introduction section, the null hypotheses of the study must be added, which must then be refuted at the end of the work in light of the results obtained
-Line 88 to better outline the work I would add a paragraph 2.1 "hydroxyapatite preparation" for example
- Line 115 the same goes for statistical analysis and for the release of substances.
-How come those hours were selected for the evaluation of the release? What is the rationale for the choice?
-A section on the limitations of the study and the possible clinical repercussions and future prospects is missing
-I noticed the presence of 7 self-citations in the reference section. For an in vitro work I think they are excessive. I ask the authors to reduce by eliminating the redundant ones

Author Response

Dear Editor,

We would like to express our sincerest gratitude to the Reviewers for their enormous efforts in criticizing the manuscript. We have taken into account all raised question here follows the detailed answers to your as well as the Reviewers. Moreover, all changes we have made to the original manuscript and marked in the red colour in the text.

Reviewer 4:

This is a very interesting study on the release of fluoride ion from nanofluorapatite
I congratulate the authors for the design and implementation of the studies.
Some criticisms are however present:

Question 2:  

An opening sentence in the abstract section on the general problem must be inserted.

Answer: An opening sentence has been added to the abstract.

Question 2:

Line 24 A final sentence in the abstract section on the possible effects and clinical implications of the study and its results should be added.

Answer: Additional information has been added to the abstract.

Question 3:

In the introduction section it is necessary to add to the possible fields of application of hydroxyapatite also the possibility, highlighted in the literature, of its use as a filler for restorative materials. In this regard, in addition to the works entered by the authors, I recommend that you include the following scientific work in the reference section, which could be of help to the reader:
Chieruzzi, M .; Pagano, S .; Lombardo, G .; Marinucci, L .; Kenny, J.M .; Torre, L .; Cianetti, S. Effect of nanohydroxyapatite, antibiotic, and mucosal defensive agent on the mechanical and thermal properties of glass ionomer cements for special needs patients. J. Mater. Res. 2018, 33, 638–649.

Answer: The highlighted scientific paper was added to the introduction section.

Question 4:

At the end of the introduction section, the null hypotheses of the study must be added, which must then be refuted at the end of the work in light of the results obtained.

Answer: The null hypothesis was added to the introduction and latter discussed in the results and conclusion section.

Question 5:

Line 88 to better outline the work I would add a paragraph 2.1 "hydroxyapatite preparation" for example.

Answer: The materials and methods section has been divided into subparagraphs for better clarity.

Question 6:

Line 115 the same goes for statistical analysis and for the release of substances.

Answer: The materials and methods section has been divided into subparagraphs for better clarity.

Question 7:

How come those hours were selected for the evaluation of the release? What is the rationale for the choice?

Answer: Currently, materials used in restorative dentistry and prophylaxis are expected to release small, long-term, continuous amounts of fluoride ions involved in enamel remineralization. Therefore, the experiment was planned to be carried out over a longer period of time (12 weeks). Multiple and frequent data readings made it possible to analyze in detail the profile of its release from the assessed FAp. A long-term study of the release of fluoride from dental materials with similar time intervals was presented in the article [34]: https://www.mdpi.com/1996-1944/14/9/2244.

Question 8:

A section on the limitations of the study and the possible clinical repercussions and future prospects is missing.

Answer: Information was added to the manuscript.

Question 9:

I noticed the presence of 7 self-citations in the reference section. For an in vitro work I think they are excessive. I ask the authors to reduce by eliminating the redundant ones.

Answer: Authors eliminated self-citations in the reference section from 7 to 5.

Round 2

Reviewer 1 Report

All the concerns of mine are solved in the revised manuscript.

Author Response

Dear Editor,

We would like once again to express our sincerest gratitude to the Reviewers for their enormous efforts in criticizing the manuscript. We have taken into account raised issue and changes that have been marked up using “track changes” function.

Reviewer 1:

All the concerns of mine are solved in the revised manuscript.

 Answer: We would like to extend our sincere gratitude for all kind assistance in the process of improving and strengthening the manuscript before its publication. We appreciate all the valid concerns raised by the Reviewer, previously.

Reviewer 2 Report

The manuscript has been much strengthened by the additional data.  I appreciate the authors' effort. 

Author Response

Dear Editor,

We would like to again express our sincerest gratitude to the Reviewers for their enormous efforts in criticizing the manuscript. We have taken into account raised issue and changes that have been marked up using “track changes” function.

Reviewer 2:

The manuscript has been much strengthened by the additional data.  I appreciate the authors' effort. 

 Answer: We would like to extend our sincere gratitude for all kind assistance in the process of improving and strengthening the manuscript before its publication. We appreciate all the valid concerns raised by the Reviewer, previously.

Reviewer 3 Report

Taking into account the improvement of the paper, I could recommend this manuscript for publication in the Ceramics journal. 

Author Response

Dear Editor,

We would like to again express our sincerest gratitude to the Reviewers for their enormous efforts in criticizing the manuscript. We have taken into account raised issue and changes that have been marked up using “track changes” function.

Reviewer 3:

Taking into account the improvement of the paper, I could recommend this manuscript for publication in the Ceramics journal.

Answer: We would like to extend our sincere gratitude for all kind assistance in the process of improving and strengthening the manuscript before its publication. We appreciate all the valid concerns raised by the Reviewer. We additionally thank the Reviewer for valuable suggestion regarding journal selection. In our humble opinion the Materials — Open Access Materials Science Journal is more suitable for publication of the manuscript entitled “In vitro assessment of long-term fluoride ion release from nanofluorapatite”. This work focuses on the assessment of long-term release of fluoride ions from nanofluorapatite in four different solutions in vitro. Therefore, it has been submitted to the Biomaterials section concerning the Influence of Nanomaterials on Biological Processes In Vitro and In Vivo. This decision is based on above stated arguments therefore we would not like to change this  journal.
